# Neural Networks with Recurrent Generative Feedback

Yujia Huang[1]   James Gornet[1]   Sihui Dai[1]   Zhiding Yu[2]   Tan Nguyen[3]
Doris Y. Tsao[1]   Anima Anandkumar[1,2]
[1]California Institute of Technology   [2]NVIDIA   [3]Rice University

## Abstract

Neural networks are vulnerable to input perturbations such as additive noise and adversarial attacks. In contrast, human perception is much more robust to such perturbations. The Bayesian brain hypothesis states that human brains use an internal generative model to update the posterior beliefs of the sensory input. This mechanism can be interpreted as a form of self-consistency between the maximum a posteriori (MAP) estimation of an internal generative model and the external environment. Inspired by such hypothesis, we enforce self-consistency in neural networks by incorporating generative recurrent feedback. We instantiate this design on convolutional neural networks (CNNs). The proposed framework, termed Convolutional Neural Networks with Feedback (CNN-F), introduces a generative feedback with latent variables to existing CNN architectures, where consistent predictions are made through alternating MAP inference under a Bayesian framework. In the experiments, CNN-F shows considerably improved adversarial robustness over conventional feedforward CNNs on standard benchmarks.

## 1   Introduction

**Vulnerability in feedforward neural networks** Conventional deep neural networks (DNNs) often contain many layers of feedforward connections. With the ever-growing network capacities and representation abilities, they have achieved great success. For example, recent convolutional neural networks (CNNs) have impressive accuracy on large scale image classification benchmarks [33]. However, current CNN models also have significant limitations. For instance, they can suffer significant performance drop from corruptions which barely influence human recognition [3]. Studies also show that CNNs can be misled by imperceptible noise known as adversarial attacks [32].

**Feedback in the human brain** To address the weaknesses of CNNs, we can take inspiration from of how human visual recognition works, and incorporate certain mecha-

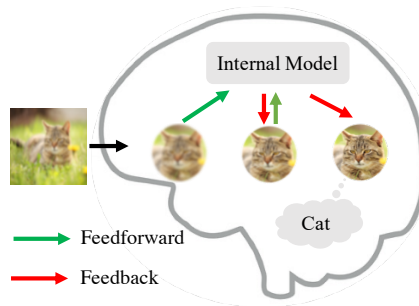

Figure 1: An intuitive illustration of recurrent generative feedback in human visual perception system.

nisms into the CNN design. While human visual cortex has hierarchical feedforward connections, backward connections from higher level to lower level cortical areas are something that current artificial networks are lacking [6]. Studies suggest these backward connections carry out top-down processing which improves the representation of sensory input [15]. In addition, evidence suggests recurrent feedback in the human visual cortex is crucial for robust object recognition. For example, humans require recurrent feedback to recognize challenging images [11]. Obfuscated images can fool humans without recurrent feedback [5]. Figure 1 shows an intuitive example of recovering a sharpened cat from a blurry cat and achieving consistent predictions after several iterations.

**Predictive coding and generative feedback** Computational neuroscientists speculate that Bayesian inference models human perception [14]. One specific formulation of predictive coding assumes Gaussian distributions on all variables and performs hierarchical Bayesian inference using recurrent, generative feedback pathways [28]. The feedback pathways encode predictions of lower level inputs, and the residual errors are used recurrently to update the predictions. In this paper, we extend the principle of predictive coding to explicitly incorporate Bayesian inference in neural networks via generative feedback connections. Specifically, we adopt a recently proposed model, named the Deconvolutional Generative Model (DGM) [25], as the generative feedback. The DGM introduces hierarchical latent variables to capture variation in images, and generates images from a coarse to fine detail using deconvolutional operations.

Our contributions are as follows:

**Self-consistency** We introduce generative feedback to neural networks and propose the self-consistency formulation for robust perception. Our internal model of the world reaches a self-consistent representation of an external stimulus. Intuitively, self-consistency says that given any two elements of label, image and auxillary information, we should be able to infer the other one. Mathematically, we use a generative model to describe the joint distribution of labels, latent variables and input image features. If the MAP estimate of each one of them are consistent with the other two, we call a label, a set of latent variables and image features to be self-consistent (Figure 4).

**CNN with Feedback (CNN-F)** We incorporate generative recurrent feedback modeled by the DGM into CNN and term this model as CNN-F. We show that Bayesian inference in the DGM is achieved by CNN with adaptive nonlinear operators (Figure 2). We impose self-consistency in the CNN-F by iterative inference and online update. Computationally, this process is done by propagating along the feedforward and feedback pathways in the CNN-F iteratively (Figure 3).

**Adversarial robustness** We show that the recurrent generative feedback in CNN-F promotes robustness and visualizes the behavior of CNN-F over iterations. We find that more iterations are needed to reach self-consistent prediction for images with larger perturbation, indicating that recurrent feedback is crucial for recognizing challenging images. When combined with adversarial training, CNN-F further improves adversarial robustness of CNN on both Fashion-MNIST and CIFAR-10 datasets.

## 2 Approach

In this section, we first formally define self-consistency. Then we give a specific form of generative feedback in CNN and impose self-consistency on it. We term this model as CNN-F. Finally we show the training and testing procedure in CNN-F. Throughout, we use the following notations:

Let $x \in \mathbb{R}^n$ be the input of a network and $y \in \mathbb{R}^K$ be the output. In image classification, $x$ is image and $y = (y^{(1)}, \ldots, y^{(K)})$ is one-hot encoded label. $K$ is the total number of classes. $K$ is usually much less than $n$. We use $L$ to denote the total number of network layers, and index the input layer to the feedforward network as layer 0. Let $h \in \mathbb{R}^m$ be encoded feature of $x$ at layer $k$ of the feedforward

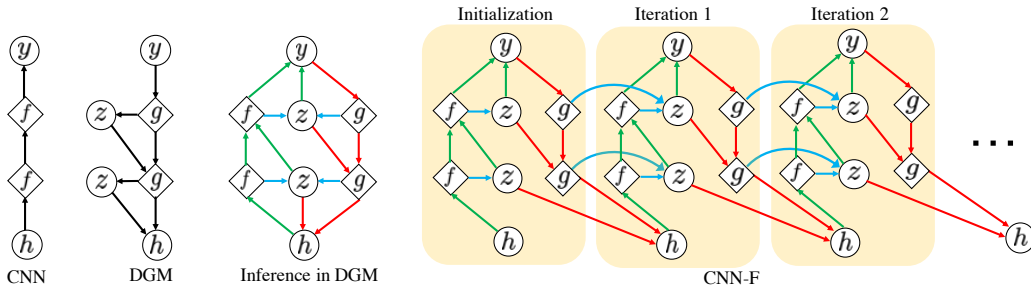

Figure 2: **Left: CNN, Graphical model for the DGM and the inference network for the DGM.** We use the DGM to as the generative model for the joint distribution of image features $h$, labels $y$ and latent variables $z$. MAP inference for $h$, $y$ and $z$ is denoted in red, green and blue respectively. $f$ and $g$ denotes feedforward features and feedback features respectively. **Right: CNN with feedback (CNN-F).** CNN-F performs alternating MAP inference via recurrent feedforward and feedback pathways to enforce self-consistency.

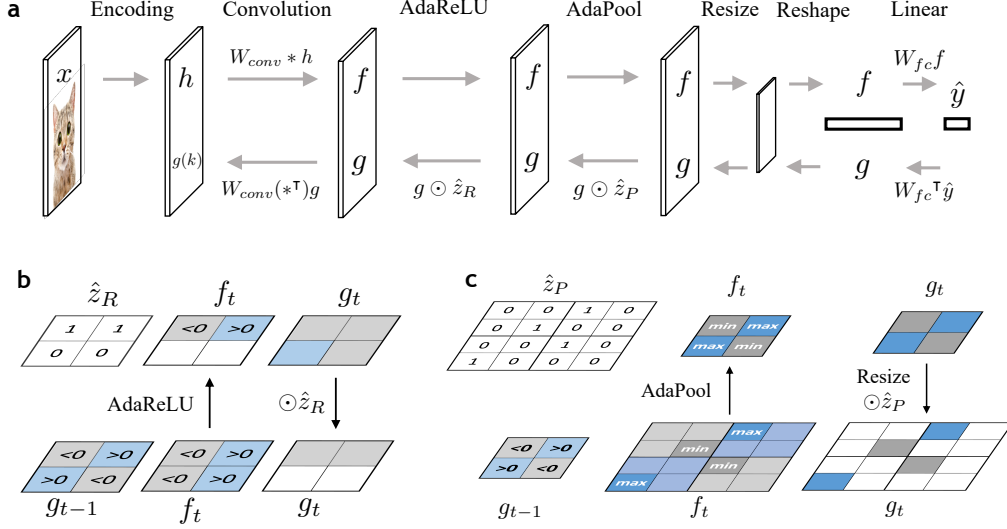

**Figure 3: Feedforward and feedback pathway in CNN-F.** a) $\hat{y}$ and $\hat{z}$ are computed by the feedforward pathway and $\hat{h}$ is computed from the feedback pathway. b) Illustration of the AdaReLU operator. c) Illustration of the AdaPool operator.

pathway. Feedforward pathway computes feature map $f(\ell)$ from layer $0$ to layer $L$, and feedback pathway generates $g(\ell)$ from layer $L$ to $k$. $g(\ell)$ and $f(\ell)$ have the same dimensions. To generate $h$ from $y$, we introduce latent variables for each layer of CNN. Let $z(\ell) \in \mathbb{R}^{C \times H \times W}$ be latent variables at layer $\ell$, where $C, H, W$ are the number of channels, height and width for the corresponding feature map. Finally, $p(h, y, z; \theta)$ denotes the joint distribution parameterized by $\theta$, where $\theta$ includes the weight $W$ and bias term $b$ of convolution and fully connected layers. We use $\hat{h}$, $\hat{y}$ and $\hat{z}$ to denote the MAP estimates of $h, y, z$ conditioning on the other two variables.

## 2.1 Generative feedback and Self-consistency

Human brain and neural networks are similar in having a hierarchical structure. In human visual perception, external stimuli are first preprocessed by lateral geniculate nucleus (LGN) and then sent to be processed by V1, V2, V4 and Inferior Temporal (IT) cortex in the ventral cortical visual system. Conventional NN use feedforward layers to model this process and learn a one-direction mapping from input to output. However, numerous studies suggest that in addition to the feedforward connections from V1 to IT, there are feedback connections among these cortical areas [6].

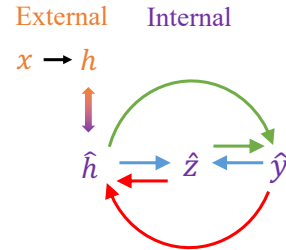

Figure 4: Self-consistency among $\hat{h}, \hat{z}, \hat{y}$ and consistency between $\hat{h}$ and $h$.

Inspired by the Bayesian brain hypothesis and the predictive coding theory, we propose to add generative feedback connections to NN. Since $h$ is usually of much higher dimension than $y$, we introduce latent variables $z$ to account for the information loss in the feedforward process. We then propose to model the feedback connections as MAP estimation from an internal generative model that describes the joint distribution of $h, z$ and $y$. Furthermore, we realize recurrent feedback by imposing self-consistency (Definition 2.1).

**Definition 2.1.** (Self-consistency) Given a joint distribution $p(h, y, z; \theta)$ parameterized by $\theta$, $(\hat{h}, \hat{y}, \hat{z})$ are self-consistent if they satisfy the following constraints:

$$\hat{y} = \arg\max_y p(y|\hat{h}, \hat{z}), \qquad \hat{h} = \arg\max_h p(h|\hat{y}, \hat{z}), \qquad \hat{z} = \arg\max_z p(z|\hat{h}, \hat{y}) \qquad (1)$$

In words, self-consistency means that MAP estimates from an internal generative model are consistent with each other. In addition to self-consistency, we also impose the consistency constraint between $\hat{h}$ and the external input features (Figure 4). We hypothesize that for *easy* images (familiar images to human, clean images in the training dataset for NN), the $\hat{y}$ from the first feedforward pass should automatically satisfy the self-consistent constraints. Therefore, feedback need not be triggered.

For *challenging* images (unfamiliar images to human, unseen perturbed images for NN), recurrent feedback is needed to obtain self-consistent $(\hat{h}, \hat{y}, \hat{z})$ and to match $\hat{h}$ with $h$. Such recurrence resembles the dynamics in neural circuits [12] and the extra effort to process challenging images [11].

## 2.2 Generative Feedback in CNN-F

CNN have been used to model the hierarchical structure of human retinatopic fields [4, 10], and have achieved state-of-the-art performance in image classification. Therefore, we introduce generative feedback to CNN and impose self-consistency on it. We term the resulting model as CNN-F.

We choose to use the DGM [25] as generative feedback in the CNN-F. The DGM introduces hierarchical binary latent variables and generates images from coarse to fine details. The generation process in the DGM is shown in Figure 3 (a). First, $y$ is sampled from the label distribution. Then each entry of $z(\ell)$ is sampled from a Bernoulli distribution parameterized by $g(\ell)$ and a bias term $b(\ell)$. $g(\ell)$ and $z(\ell)$ are then used to generate the layer below:

$$g(\ell - 1) = W(*^{\intercal})(\ell)(z(\ell) \odot g(\ell)) \tag{2}$$

In this paper, we assume $p(y)$ to be uniform, which is realistic under the balanced label scenario. We assume that $h$ follows Gaussian distribution centered at $g(k)$ with standard deviation $\sigma$.

## 2.3 Recurrence in CNN-F

In this section, we show that self-consistent $(\hat{h}, \hat{y}, \hat{z})$ in the DGM can be obtained via alternately propagating along feedforward and feedback pathway in CNN-F.

**Feedforward and feedback pathway in CNN-F**    The feedback pathway in CNN-F takes the same form as the generation process in the DGM (Equation (2)). The feedforward pathway in CNN-F takes the same form as CNN except for the nonlinear operators. In conventional CNN, nonlinear operators are $\sigma_{\text{ReLU}}(f) = \max(f, 0)$ and $\sigma_{\text{MaxPool}}(f) = \max_{r \times r} f$, where $r$ is the dimension of the pooling region in the feature map (typically equals to 2 or 3). In contrast, we use $\sigma_{\text{AdaReLU}}$ and $\sigma_{\text{AdaPool}}$ given in Equation (3) in the feedforward pathway of CNN-F. These operators adaptively choose how to activate the feedforward feature map based on the sign of the feedback feature map. The feedforward pathway computes $f(\ell)$ using the recursion $f(\ell) = W(\ell) * \sigma(f(\ell - 1)) \} + b(\ell)$ [1].

$$\sigma_{\text{AdaReLU}}(f) = \begin{cases} \sigma_{\text{ReLU}}(f), & \text{if } g \geq 0 \\ \sigma_{\text{ReLU}}(-f), & \text{if } g < 0 \end{cases} \qquad \sigma_{\text{AdaPool}}(f) = \begin{cases} \sigma_{\text{MaxPool}}(f), & \text{if } g \geq 0 \\ -\sigma_{\text{MaxPool}}(-f), & \text{if } g < 0 \end{cases} \tag{3}$$

**MAP inference in the DGM**    Given a joint distribution of $h, y, z$ modeled by the DGM, we aim to show that we can make predictions using a CNN architecture following the Bayes rule (Theorem 2.1). To see this, first recall that generative classifiers learn a joint distribution $p(x, y)$ of input data $x$ and their labels $y$, and make predictions by computing $p(y|x)$ using the Bayes rule. A well known example is the Gaussian Naive Bayes model (GNB). The GNB models $p(x, y)$ by $p(y)p(x|y)$, where $y$ is Boolean variable following a Bernoulli distribution and $p(x|y)$ follows Gaussian distribution. It can be shown that $p(y|x)$ computed from GNB has the same parametric form as logistic regression.

**Assumption 2.1.** (Constancy assumption in the DGM).

**A.** The generated image $g(k)$ at layer $k$ of DGM satisfies $||g(k)||_2^2 = $ const.
**B.** Prior distribution on the label is a uniform distribution: $p(y) = $ const.
**C.** Normalization factor in $p(z|y)$ for each category is constant: $\sum_z e^{\eta(y,z)} = $ const.

**Remark.** To meet Assumption 2.1.A, we can normalize $g(k)$ for all $k$. This results in a form similar to the instance normalization that is widely used in image stylization [35]. See Appendix A.4 for more detailed discussion. Assumption 2.1.B assumes that the label distribution is balanced. $\eta$ in Assumption 2.1.C is used to parameterize $p(z|y)$. See Appendix A for the detailed form.

**Theorem 2.1.** Under Assumption 2.1 and given a joint distribution $p(h, y, z)$ modeled by the DGM, $p(y|h, z)$ has the same parametric form as a CNN with $\sigma_{\text{AdaReLU}}$ and $\sigma_{\text{AdaPool}}$.

*Proof.* Please refer to Appendix A. □

**Remark.** Theorem 2.1 says that DGM and CNN is a generative-discriminative pair in analogy to GNB and logistic regression.

We also find the form of MAP inference for image feature $\hat{h}$ and latent variables $\hat{z}$ in the DGM. Specifically, we use $z_R$ and $z_P$ to denote latent variables that are at a layer followed by $\mathrm{AdaReLU}$ and $\mathrm{AdaPool}$ respectively. $\mathbb{1}(\cdot)$ denotes indicator function.

**Proposition 2.1** (MAP inference in the DGM). Under Assumption 2.1, the following hold:

   **A.** Let $h$ be the feature at layer $k$, then $\hat{h} = g(k)$.
   **B.** MAP estimate of $z(\ell)$ conditioned on $h, y$ and $\{z(j)\}_{j \neq \ell}$ in the DGM is:
$$\hat{z}_R(\ell) = \mathbb{1}(\sigma_{\mathrm{AdaReLU}}(f(\ell)) \geq 0) \tag{4}$$
$$\hat{z}_P(\ell) = \mathbb{1}(g(\ell) \geq 0) \odot \arg\max_{r \times r}(f(\ell)) + \mathbb{1}(g(\ell) < 0) \odot \arg\min_{r \times r}(f(\ell)) \tag{5}$$

*Proof.* For part A, we have $\hat{h} = \arg\max_h p(h|\hat{y}, \hat{z}) = \arg\max_h p(h|g(k)) = g(k)$. The second equality is obtained because $g(k)$ is a deterministic function of $\hat{y}$ and $\hat{z}$. The third equality is obtained because $h \sim \mathcal{N}(g(k), \mathrm{diag}(\sigma^2))$. For part B, please refer to Appendix A. $\square$

**Remark.** Proposition 2.1.A show that $\hat{h}$ is the output of the generative feedback in the CNN-F. Proposition 2.1.B says that $\hat{z}_R = 1$ if the sign of the feedforward feature map matches with that of the feedback feature map. $\hat{z}_P = 1$ at locations that satisfy one of these two requirements: 1) the value in the feedback feature map is non-negative and it is the maximum value within the local pooling region or 2) the value in the feedback feature map is negative and it is the minimum value within the local pooling region. Using Proposition 2.1.B, we approximate $\{\hat{z}(\ell)\}_{\ell=1:L}$ by greedily finding the MAP estimate of $\hat{z}(\ell)$ conditioning on all other layers.

**Iterative inference and online update in CNN-F**  We find self-consistent $(\hat{h}, \hat{y}, \hat{z})$ by iterative inference and online update (Algorithm 1). In the initialization step, image $x$ is first encoded to $h$ by $k$ convolutional layers. Then $h$ passes through a standard CNN, and latent variables are initialized with conventional $\sigma_{\mathrm{ReLU}}$ and $\sigma_{\mathrm{MaxPool}}$. The feedback generative network then uses $\hat{y}_0$ and $\{\hat{z}_0(\ell)\}_{\ell=k:L}$ to generate intermediate features $\{g_0(\ell)\}_{\ell=k:L}$, where the subscript denotes the number of iterations. In practice, we use logits instead of one-hot encoded label in the generative feedback to maintain uncertainty in each category. We use $g_0(k)$ as the input features for the first iteration. Starting from this iteration, we use $\sigma_{\mathrm{AdaReLU}}$ and $\sigma_{\mathrm{AdaPool}}$ instead of $\sigma_{\mathrm{ReLU}}$ and and $\sigma_{\mathrm{MaxPool}}$ in the feedforward pathway to infer $\hat{z}$ (Equation (20) and (21)). In practice, we find that instead of greedily replacing the input with generated features and starting a new inference iteration, online update eases the training and gives better robustness performance. The online update rule of CNN-F can be written as:
$$\hat{h}_{t+1} \leftarrow \hat{h}_t + \eta(g_{t+1}(k) - \hat{h}_t) \tag{6}$$
$$f_{t+1}(\ell) \leftarrow f_{t+1}(\ell) + \eta(g_t(\ell) - f_{t+1}(\ell)), \ell = k, \ldots, L \tag{7}$$
where $\eta$ is the step size. Greedily replacement is a special case for the online update rule when $\eta = 1$.

---

**Algorithm 1:** Iterative inference and online update in CNN-F

**Input :** Input image $x$, number of encoding layers $k$, maximum number of iterations N.
Encode image $x$ to $h_0$ with $k$ convolutional layers;
Initialize $\{\hat{z}(\ell)\}_{\ell=k:L}$ by $\sigma_{\mathrm{ReLU}}$ and $\sigma_{\mathrm{MaxPool}}$ in the standard CNN;
**while** $t < N$ **do**

    Feedback pathway: generate $g_t(k)$ using $\hat{y}_t$ and $\hat{z}_t(\ell)$, $\ell = k, \ldots, L$;

    Feedforward pathway: use $\hat{h}_{t+1}$ as the input (Equation (6));
                         update each feedforward layer using Equation (7);
                         predict $\hat{y}_{t+1}$ using the updated feedforward layers;

**end**
**return** $\hat{h}_N, \hat{y}_N, \hat{z}_N$

---

## 2.4  Training the CNN-F

During training, we have three goals: 1) train a generative model to model the data distribution, 2) train a generative classifier and 3) enforce self-consistency in the model. We first approximate self-consistent $(\hat{h}, \hat{y}, \hat{z})$ and then update model parameters based on the losses listed in Table 1. All losses are computed for every iteration. Minimizing the reconstruction loss increases data likelihood given current estimates of label and latent variables $\log p(h|\hat{y}_t, \hat{z}_t)$ and enforces consistency between

$\hat{h}_t$ and $h$. Minimizing the cross-entropy loss helps with the classification goal. In addition to reconstruction loss at the input layer, we also add reconstruction loss between intermediate feedback and feedforward feature maps. These intermediate losses helps stabilizing the gradients when training an iterative model like the CNN-F.

Table 1: Training losses in the CNN-F.

|  | Form | Purpose |
| --- | --- | --- |
| Cross-entropy loss | $\log p(y \mid \hat{h}_t, \hat{z}_t; \theta)$ | classification |
| Reconstruction loss | $\log p(h \mid \hat{y}_t, \hat{z}_t; \theta) = \|h - \hat{h}\|_2^2$ | generation, self-consistency |
| Intermediate reconstruction loss | $\|f_0(\ell) - g_t(\ell)\|_2^2$ | stabilizing training |

## 3 Experiment

### 3.1 Generative feedback promotes robustness

As a sanity check, we train a CNN-F model with two convolution layers and one fully-connected layer on clean Fashion-MNIST images. We expect that CNN-F reconstructs the perturbed inputs to their clean version and makes self-consistent predictions. To this end, we verify the hypothesis by evaluating adversarial robustness of CNN-F and visualizing the restored images over iterations.

**Adversarial robustness** Since CNN-F is an iterative model, we consider two attack methods: attacking the first or last output from the feedforward streams. We use "first" and "e2e" (short for end-to-end) to refer to the above two attack approaches, respectively. Due to the approximation of non-differentiable activation operators and the depth of the unrolled CNN-F, end-to-end attack is weaker than first attack (Appendix B.1). We report the adversarial accuracy against the stronger attack in Figure 5. We use the Fast Gradient Sign Attack Method (FGSM) [8] Projected Gradient Descent (PGD) method to attack. For PGD attack, we generate adversarial samples within $L_\infty$-norm constraint, and denote the maximum $L_\infty$-norm between adversarial images and clean images as $\epsilon$.

Figure 5 (a, b) shows that the CNN-F improves adversarial robustness of a CNN on Fashion-MNIST without access to adversarial images during training. The error bar shows standard deviation of 5 runs. Figure 5 (c) shows that training a CNN-F with more iterations improves robustness. Figure 5 (d) shows that the predictions are corrected over iterations during testing time for a CNN-F trained with 5 iterations. Furthermore, we see larger improvements for higher $\epsilon$. This indicates that recurrent feedback is crucial for recognizing challenging images.

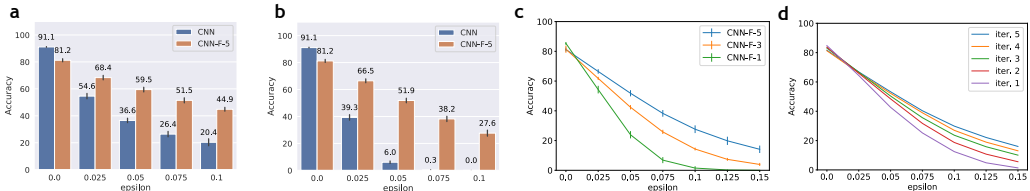

Figure 5: **Adversarial robustness of CNN-F with standard training on Fashion-MNIST.** CNN-F-$k$ stands for CNN-F trained with $k$ iterations. a) Attack with FGSM. b) Attack with PGD using 40 steps. c) Train with different number of iterations. Attack with PGD-40. d) Evaluate a trained CNN-F-5 model with various number of iterations against PGD-40 attack.

**Image restoration** Given that CNN-F models are robust to adversarial attacks, we examine the models' mechanism for robustness by visualizing how the generative feedback moves a perturbed image over iterations. We select a validation image from Fashion-MNIST. Using the image's two largest principal components, a two-dimensional hyperplane $\subset \mathbb{R}^{28 \times 28}$ intersects the image with the image at the center. Vector arrows visualize the generative feedback's movement on the hyperplane's position. In Figure 6 (a), we find that generative feedback perturbs samples across decision boundaries toward the validation image. This demonstrates that the CNN-F's generative feedback can restore perturbed images to their uncorrupted objects.

We further explore this principle with regard to adversarial examples. The CNN-F model can correct initially wrong predictions. Figure 6 (b) uses Grad-CAM activations to visualize the network's

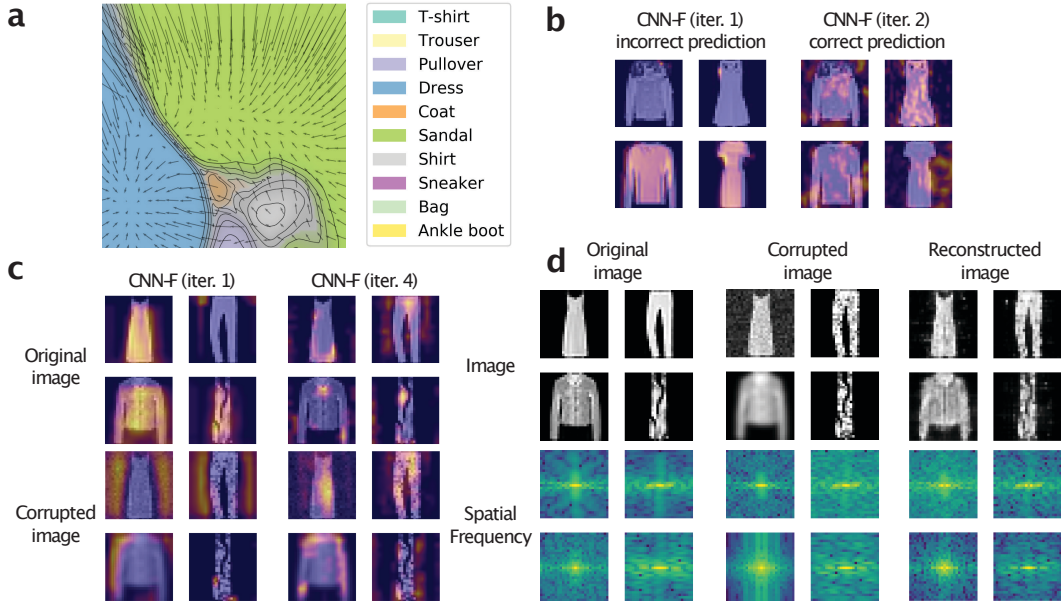

Figure 6: **The generative feedback in CNN-F models restores perturbed images.** a) The decision cell cross-sections for a CNN-F trained on Fashion-MNIST. Arrows visualize the feedback direction on the cross-section. b) Fashion-MNIST classification accuracy on PGD adversarial examples; Grad-CAM activations visualize the CNN-F model's attention from incorrect (iter. 1) to correct predictions (iter. 2). c) Grad-CAM activations across different feedback iterations in the CNN-F. d) From left to right: clean images, corrupted images, and images restored by the CNN-F's feedback.

attention from an incorrect prediction to a correct prediction on PGD-40 adversarial samples [30]. To correct predictions, the CNN-F model does not initially focus on specific features. Rather, it either identifies the entire object or the entire image. With generative feedback, the CNN-F begins to focus on specific features. This is reproduced in clean images as well as images corrupted by blurring and additive noise 6 (c). Furthermore, with these perceptible corruptions, the CNN-F model can reconstruct the clean image with generative feedback 6 (d). This demonstrates that the generative feedback is one mechanism that restores perturbed images.

## 3.2   Adversarial Training

Adversarial training is a well established method to improve adversarial robustness of a neural network [20]. Adversarial training often solves a minimax optimization problem where the attacker aims to maximize the loss and the model parameters aims to minimize the loss. In this section, we show that CNN-F can be combined with adversarial training to further improve the adversarial robustness.

**Training methods**   Figure 7 illustrates the loss design we use for CNN-F adversarial training. Different from standard adversarial training on CNNs, we use cross-entropy loss on both clean images and adversarial images. In addition, we add reconstruction loss between generated features of adversarial samples from iterative feedback and the features of clean images in the first forward pass.

**Experimental setup**   We train the CNN-F on Fashion-MNIST and CIFAR-10 datasets respectively. For Fashion-MNIST, we train a network with 4 convolution layers and 3 fully-connected layers. We

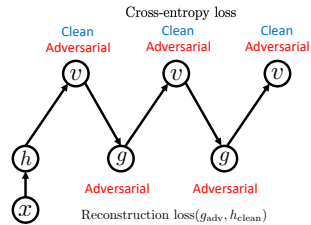

Figure 7: **Loss design for CNN-F adversarial training**, where $v$ stands for the logits. $x$, $h$ and $g$ are input image, encoded feature, and generated feature, respectively.

use 2 convolutional layers to encode the image into feature space and reconstruct to that feature space. For CIFAR-10, we use the WideResNet architecture [39] with depth 40 and width 2. We reconstruct to the feature space after 5 basic blocks in the first network block. For more detailed hyper-parameter settings, please refer to Appendix B.2. During training, we use PGD-7 to attack the first forward pass of CNN-F to obtain adversarial samples. During testing, we also perform SPSA

[34] and transfer attack in addition to PGD attack to prevent the gradient obfuscation [1] issue when evaluating adversarial robustness of a model. In the transfer attack, we use the adversarial samples of the CNN to attack CNN-F.

**Main results** CNN-F further improves the robustness of CNN when combined with adversarial training. Table 2 and Table 3 list the adversarial accuracy of CNN-F against several attack methods on Fashion-MNIST and CIFAR-10. On Fashion-MNIST, we train the CNN-F with 1 iterations. On CIFAR-10, we train the CNN-F with 2 iterations. We report two evaluation methods for CNN-F: taking the logits from the last iteration (last), or taking the average of logits from all the iterations (avg). We also report the lowest accuracy among all the attack methods with bold font to highlight the weak spot of each model. In general, we find that the CNN-F tends to be more robust to end-to-end attack compared with attacking the first forward pass. This corresponds to the scenario where the attacker does not have access to internal iterations of the CNN-F. Based on different attack scenarios, we can tune the hyper-paramters and choose whether averaging the logits or outputting the logits from the last iteration to get the best robustness performance (Appendix B.2).

Table 2: Adversarial accuracy on Fashion-MNIST over 3 runs. $\epsilon = 0.1$.

|  | Clean | PGD (first) | PGD (e2e) | SPSA (first) | SPSA (e2e) | Transfer | Min |
|---|---|---|---|---|---|---|---|
| CNN | $\mathbf{89.97 \pm 0.10}$ | $77.09 \pm 0.19$ | $77.09 \pm 0.19$ | $87.33 \pm 1.14$ | $87.33 \pm 1.14$ | — | $77.09 \pm 0.19$ |
| CNN-F (last) | $89.87 \pm 0.14$ | $79.19 \pm 0.49$ | $78.34 \pm 0.29$ | $87.10 \pm 0.10$ | $87.33 \pm 0.89$ | $82.76 \pm 0.26$ | $78.34 \pm 0.29$ |
| CNN-F (avg) | $89.77 \pm 0.08$ | $\mathbf{79.55 \pm 0.15}$ | $\mathbf{79.89 \pm 0.16}$ | $\mathbf{88.27 \pm 0.91}$ | $\mathbf{88.23 \pm 0.81}$ | $\mathbf{83.15 \pm 0.17}$ | $\mathbf{79.55 \pm 0.15}$ |

Table 3: Adversarial accuracy on CIFAR-10 over 3 runs. $\epsilon = 8/255$.

|  | Clean | PGD (first) | PGD (e2e) | SPSA (first) | SPSA (e2e) | Transfer | Min |
|---|---|---|---|---|---|---|---|
| CNN | $79.09 \pm 0.11$ | $42.31 \pm 0.51$ | $42.31 \pm 0.51$ | $66.61 \pm 0.09$ | $66.61 \pm 0.09$ | — | $42.31 \pm 0.51$ |
| CNN-F (last) | $78.68 \pm 1.33$ | $\mathbf{48.90 \pm 1.30}$ | $49.35 \pm 2.55$ | $68.75 \pm 1.90$ | $51.46 \pm 3.22$ | $66.19 \pm 1.37$ | $\mathbf{48.90 \pm 1.30}$ |
| CNN-F (avg) | $\mathbf{80.27 \pm 0.69}$ | $48.72 \pm 0.64$ | $\mathbf{55.02 \pm 1.91}$ | $\mathbf{71.56 \pm 2.03}$ | $\mathbf{58.83 \pm 3.72}$ | $\mathbf{67.09 \pm 0.68}$ | $48.72 \pm 0.64$ |

# 4 Related work

**Robust neural networks with latent variables** Latent variable models are a unifying theme in robust neural networks. The consciousness prior [2] postulates that natural representations—such as language—operate in a low-dimensional space, which may restrict expressivity but also may facilitate rapid learning. If adversarial attack introduce examples outside this low-dimensional manifold, latent variable models can map these samples back to the manifold. A related mechanism for robustness is state reification [17]. Similar to self-consistency, state reification models the distribution of hidden states over the training data. It then maps less likely states to more likely states. MagNet and Denoising Feature Matching introduce similar mechanisms: using autoencoders on the input space to detect adversarial examples and restore them in the input space [21, 37]. Lastly, Defense-GAN proposes a generative adversarial network to approximate the data manifold [29]. CNN-F generalizes these themes into a Bayesian framework. Intuitively, CNN-F can be viewed as an autoencoder. In contrast to standard autoencoders, CNN-F requires stronger constraints through Bayes rule. CNN-F—through self-consistency—constrains the generated image to satisfy the *maximum a posteriori* on the predicted output.

**Computational models of human vision** Recurrent models and Bayesian inference have been two prevalent concepts in computational visual neuroscience. Recently, Kubilius et al. [16] proposed CORnet as a more accurate model of human vision by modeling recurrent cortical pathways. Like CNN-F, they show CORnet has a larger V4 and IT neural similarity compared to a CNN with similar weights. Linsley et al. [19] suggests hGRU as another recurrent model of vision. Distinct from other models, hGRU models lateral pathways in the visual cortex to global contextual information. While Bayesian inference is a candidate for visual perception, a Bayesian framework is absent in these models. The recursive cortical network (RCN) proposes a hierarchal conditional random field as a model for visual perception [7]. In contrast to neural networks, RCN uses belief propagation for both training and inference. With the representational ability of neural networks, we propose CNN-F to approximate Bayesian inference with recurrent circuits in neural networks.

**Feedback networks** Feedback Network [40] uses convLSTM as building blocks and adds skip connections between different time steps. This architecture enables early prediction and enforces

hierarchical structure in the label space. Nayebi et al. [24] uses architecture search to design local recurrent cells and long range feedback to boost classification accuracy. Wen et al. [38] designs a bi-directional recurrent neural network by recursively performing bottom up and top down computations. The model achieves more accurate and definitive image classification. In addition to standard image classification, neural networks with feedback have been applied to other settings. Wang et al. [36] propose a feedback-based propagation approach that improves inference in CNN under partial evidence in the multi-label setting. Piekniewski et al. [27] apply multi-layer perceptrons with lateral and feedback connections to visual object tracking.

**Combining top-down and bottom-up signals in RNNs** Mittal et al. [23] proposes combining attention and modularity mechanisms to route bottom-up (feedforward) and top-down (feedback) signals. They extend the Recurrent Independent Mechanisms (RIMs) [9] framework to a bidirectional structure such that each layer of the hierarchy can send information in both bottom-up direction and top-down direction. Our approach uses approximate Bayesian inference to provide top-down communication, which is more consistent with the Bayesian brain framework and predictive coding.

**Inference in generative classifiers** Sulam et al. [31] derives a generative classifier using a sparse prior on the layer-wise representations. The inference is solved by a multi-layer basis pursuit algorithm, which can be implemented via recurrent convolutional neural networks. Nimmagadda and Anandkumar [26] propose to learn a latent tree model in the last layer for multi-object classification. A tree model allows for one-shot inference in contrast to iterative inference.

**Target propagation** The generative feedback in CNN-F shares a similar form as target propagation, where the targets at each layer are propagated backwards. In addition, difference target propagation uses auto-encoder like losses at intermediate layers to promote network invertibility [22, 18]. In the CNN-F, the intermediate reconstruction loss between adversarial and clean feature maps during adversarial training promotes the feedback to project perturbed image back to its clean version in all resolution scales.

## 5 Conclusion

Inspired by the recent studies in Bayesian brain hypothesis, we propose to introduce recurrent generative feedback to neural networks. We instantiate the framework on CNN and term the model as CNN-F. In the experiments, we demonstrate that the proposed feedback mechanism can considerably improve the adversarial robustness compared to conventional feedforward CNNs. We visualize the dynamical behavior of CNN-F and show its capability of restoring corrupted images. Our study shows that the generative feedback in CNN-F presents a biologically inspired architectural design that encodes inductive biases to benefit network robustness.

## Broader Impacts

Convolutional neural networks (CNNs) can achieve superhuman performance on image classification tasks. This advantage allows their deployment to computer vision applications such as medical imaging, security, and autonomous driving. However, CNNs trained on natural images tend to overfit to image textures. Such flaw can cause a CNN to fail against adversarial attacks and on distorted images. This may further lead to unreliable predictions potentially causing false medical diagnoses, traffic accidents, and false identification of criminal suspects. To address the robustness issues in CNNs, CNN-F adopts an architectural design which resembles human vision mechanisms in certain aspects. The deployment of CNN-F renders more robust AI systems.

Despite the improved robustness, current method does not tackle other social and ethical issues intrinsic to a CNN. A CNN can imitate human biases in the image datasets. In automated surveillance, biased training datasets can improperly calibrate CNN-F systems to make incorrect decisions based on race, gender, and age. Furthermore, while robust, human-like computer vision systems can provide a net positive societal impact, there exists potential use cases with nefarious, unethical purposes. More human-like computer vision algorithms, for example, could circumvent human verification software. Motivated by these limitations, we encourage research into human bias in machine learning and security in computer vision algorithms. We also recommend researchers and policymakers examine how people abuse CNN models and mitigate their exploitation.

## Acknowledgements

We thank Chaowei Xiao, Haotao Wang, Jean Kossaifi, Francisco Luongo for the valuable feedback. Y. Huang is supported by DARPA LwLL grants. J. Gornet is supported by supported by the NIH Predoctoral Training in Quantitative Neuroscience 1T32NS105595-01A1. D. Y. Tsao is supported by Howard Hughes Medical Institute and Tianqiao and Chrissy Chen Institute for Neuroscience. A. Anandkumar is supported in part by Bren endowed chair, DARPA LwLL grants, Tianqiao and Chrissy Chen Institute for Neuroscience, Microsoft, Google, and Adobe faculty fellowships.

## Footnotes

[1] $\sigma$ takes the form of $\sigma_{\text{AdaPool}}$ or $\sigma_{\text{AdaReLU}}$.

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
