[Supplementary Material]

# Appendix

## A   Inference in the Deconvolutional Generative Model

### A.1   Generative model

We choose the deconvolutional generative model (DGM) [25] as the generative feedback in CNN-F. The graphical model of the DGM is shown in Figure 2 (middle). The DGM has the same architecture as CNN and generates images from high level to low level. Since low level features usually have higher dimension than high level features, the DGM introduces latent variables at each level to account for uncertainty in the generation process.

Let $y \in \mathbb{R}^K$ be label, $K$ is the number of classes. Let $x \in \mathbb{R}^n$ be image and $h \in \mathbb{R}^m$ be encoded features of $x$ after $k$ convolutional layers. In a DGM with $L$ layers in total, $g(\ell) \in \mathbb{R}^{C \times H \times W}$ denotes generated feature map at layer $\ell$, and $z(\ell) \in \mathbb{R}^{C \times H \times W}$ denotes latent variables at layer $\ell$. We use $z_R$ and $z_P$ to denote latent variables at a layer followed by ReLU and MaxPool respectively. In addition, we use $(\cdot)^{(i)}$ to denote the $i$th entry in a tensor. Let $W(\ell)$ and $b(\ell)$ be the weight and bias parameters at layer $\ell$ in the DGM. We use $(\cdot)(*^\intercal)$ to denote deconvolutional transpose in deconvolutional layers and $(\cdot)^\intercal$ to denote matrix transpose in fully connected layers. In addition, we use $(\cdot)_\uparrow$ and $(\cdot)_\downarrow$ to denote upsampling and downsampling. The generation process in the DGM is as follows:

$$y \sim p(y) \tag{8}$$

$$g(L-1) = W(L)^\intercal y \tag{9}$$

$$z_P(L-1)^{(i)} \sim \text{Ber}\left( \frac{e^{b(L-1) \cdot g(L-1)_\uparrow^{(i)}}}{e^{b(L-1) \cdot g(L-1)_\uparrow^{(i)}} + 1} \right) \tag{10}$$

$$g(L-2) = W(L-1)(*^\intercal)\{g(L-1)_\uparrow \odot z_P(L-1)\} \tag{11}$$

$$\vdots$$

$$z_R(\ell)^{(i)} \sim \text{Ber}\left( \frac{e^{b(\ell) \cdot g(\ell)^{(i)}}}{e^{b(\ell) \cdot g(\ell)^{(i)}} + 1} \right) \tag{12}$$

$$g(\ell-1) = W(\ell)(*^\intercal)\{z_R(\ell) \odot g(\ell)\} \tag{13}$$

$$\vdots$$

$$x \sim \mathcal{N}(g(0), \text{diag}(\sigma^2)) \tag{14}$$

In the above generation process, we generate all the way to the image level. If we choose to stop at layer $k$ to generate image features $h$, the final generation step is $h \sim \mathcal{N}(g(k), \text{diag}(\sigma^2))$ instead of (14). The joint distribution of latent variables from layer 1 to $L$ conditioning on $y$ is:

$$p(\{z(\ell)\}_{\ell=1:L}|y) = p(z(L)|y)\Pi_{\ell=1}^{L-1}p(z(\ell)|\{z(k)\}_{k \geq \ell}, y)$$

$$= \text{Softmax}\left( \sum_{\ell=1}^{L} \langle b(\ell), z(\ell) \odot g(\ell)\rangle \right) \tag{15}$$

where $\text{Softmax}(\eta) = \frac{\exp(\eta)}{\sum_\eta \exp(\eta)}$ with $\eta = \sum_{\ell=1}^{L} \langle b(\ell), z(\ell) \odot g(\ell)\rangle$.

### A.2   Proof for Theorem 2.1

In this section, we provide proofs for Theorem 2.1. In the proof, we use $f$ to denote the feedforward feature map after convolutional layer in the CNN of the same architecture as the DGM, and use $(\cdot)_a$ to denote layers after nonlinear operators. Let $v$ be the logits output from fully-connected layer of the CNN. Without loss of generality, we consider a DGM that has the following architecture. We list the corresponding feedforward feature maps on the left column:

$$\text{Conv} \quad f(1) = W(1) * x + b(1) \qquad\qquad g(0) = W(1)(*^\intercal)g_a(1)$$
$$\text{ReLU} \quad f_a(1) = \sigma_{\text{AdaReLU}}(f(1)) \qquad\qquad g_a(1) = g(1) \odot z_R(1)$$
$$\text{Conv} \quad f(2) = W(2) * f_a(1) + b(2) \qquad\qquad g(1) = W(2)(*^\intercal)g_a(2)$$
$$\text{Pooling} \quad f_a(2) = \sigma_{\text{AdaPool}}(f(2)) \qquad\qquad g_a(2) = g(2)_\uparrow \odot z_P(2)$$
$$\text{FC} \quad v = W(3)f_a(2) \qquad\qquad g(2) = W(3)^\intercal v$$

We prove Theorem 2.1 which states that CNN with $\sigma_{\text{AdaReLU}}$ and $\sigma_{\text{AdaPool}}$ is the generative classifier derived from the DGM by proving Lemma A.1 first.

**Definition A.1.** $\sigma_{\text{AdaReLU}}$ and $\sigma_{\text{AdaPool}}$ are nonlinear operators that adaptively choose how to activate the feedforward feature map based on the sign of the feedback feature map.

$$\sigma_{\text{AdaReLU}}(f) = \begin{cases} \sigma_{\text{ReLU}}(f), & \text{if } g \geq 0 \\ \sigma_{\text{ReLU}}(-f), & \text{if } g < 0 \end{cases} \qquad \sigma_{\text{AdaPool}}(f) = \begin{cases} \sigma_{\text{MaxPool}}(f), & \text{if } g \geq 0 \\ -\sigma_{\text{MaxPool}}(-f), & \text{if } g < 0 \end{cases} \qquad (16)$$

**Definition A.2** (generative classifier). Let $v$ be the logits output of a CNN, and $p(x, y, z)$ be the joint distribution specified by a generative model. A CNN is a generative classifier of a generative model if $\text{Softmax}(v) = p(y|x, z)$.

**Lemma A.1.** *Let $y$ be the label and $x$ be the image. $v$ is the logits output of the CNN that has the same architecture and parameters as the DGM. $g(0)$ is the generated image from the DGM. $\alpha$ is a constant. $\eta(y, z) = \sum_{\ell=1}^{L} \langle b(\ell), z(\ell) \odot g(\ell) \rangle$. Then we have:*

$$\alpha y^\intercal v = g(0)^\intercal x + \eta(y, z) \qquad (17)$$

*Proof.*

$$g(0)^\intercal x + \eta(y, z)$$
$$= \{W(1)(*^\intercal)\{g(1) \odot z_R(1)\}\}^\intercal x + (z_R(1) \odot g(1))^\intercal b(1) + (z_P(2) \odot g(2)_\uparrow)^\intercal b(2)$$
$$= (z_R(1) \odot g(1))^\intercal \{W(1)(*^\intercal)x + b(1)\} + (z_P(2) \odot g(2)_\uparrow)^\intercal b(2)$$
$$= g(1)^\intercal (z_R(1) \odot f(1)) + (z_P(2) \odot g(2)_\uparrow)^\intercal b(2)$$
$$= \{W(2)(*^\intercal)\{g(2)_\uparrow \odot z_P(2)\}\}^\intercal (z_R(1) \odot f(1)) + (z_P(2) \odot g(2)_\uparrow)^\intercal b(2)$$
$$= \{g(2)_\uparrow \odot z_P(2)\}^\intercal \{W(2) * (z_R(1) \odot f(1)) + b(2)\}$$
$$= (W(3)^\intercal y)_\uparrow^\intercal \{z_P(2) \odot f(2)\}$$
$$= \alpha(W(3)^\intercal y)^\intercal (z_P(2) \odot f(2))_\downarrow$$
$$= \alpha y^\intercal W(3)(z_P(2) \odot f(2))_\downarrow$$
$$= \alpha y^\intercal v$$

$\square$

**Remark.** *Lemma A.1 shows that logits output from the corresponding CNN of the DGM is proportional to the inner product of generated image and input image plus $\eta(y, z)$. Recall from Equation (14), since the DGM assumes $x$ to follow a Gaussian distribution centered at $g(0)$, the inner product between $g(0)$ and $x$ is related to $\log p(x|y, z)$. Recall from Equation (15) that conditionoal distribution of latent variables in the DGM is parameterized by $\eta(y, z)$. Using these insights, we can use Lemma A.1 to show that CNN performs Bayesian inference in the DGM.*
*In the proof, the fully-connected layer applies a linear transformation to the input without any bias added. For fully-connected layer with bias term, we modify $\eta(y, z)$ to $\eta'(y, z)$:*

$$\eta'(y, z) = \eta(y, z) + y^\intercal b(3)$$

*The logits are computed by*

$$v = W(3)(f(2) \odot z(2)) + b(3)$$

*Following a very similar proof as of Lemma A.1, we can show that*

$$\alpha y^\intercal v = g^\intercal(0) + \eta'(y, z) \qquad (18)$$

With Lemma A.1, we can prove Theorem 2.1. Here, we repeat the theorem and the assumptions on which it is defined:

**Assumption 2.1.** (Constancy assumption in the DGM)

**A.** The generated image $g(k)$ at layer $k$ of DGM satisfies $||g(k)||_2^2 = $ const.
**B.** Prior distribution on the label is a uniform distribution: $p(y) = $ const.
**C.** Normalization factor in $p(z|y)$ for each category is constant: $\sum_z e^{\eta(y,z)} = $ const.

**Theorem 2.1.** Under Assumption 2.1, and given a joint distribution $p(h, y, z)$ modeled by the DGM, $p(y|h, z)$ has the same parametric form as a CNN with $\sigma_{\text{AdaReLU}}$ and $\sigma_{\text{AdaPool}}$.

*Proof.* Without loss of generality, assume that we generate images at a pixel level. In this case, $h = x$. We use $p(x, y, z)$ to denote the joint distribution specified by the DGM. In addition, we use $q(y|x, z)$ to denote the Softmax output from the CNN, i.e. $q(y|x, z) = \frac{y^\intercal e^v}{\sum_{i=1}^{K} e^{v^{(i)}}}$. To simplify the notation, we use $z$ instead of $\{z(\ell)\}_{\ell=1:L}$ to denote latent variables across layers.

$$
\begin{aligned}
&\log p(y|x, z) \\
=& \log p(y, x, z) - \log p(x, z) \\
=& \log p(x|y, z) + \log p(z|y) + \log p(y) - \log p(x, z) \\
=& \log p(x|y, z) + \log p(z|y) + \text{const.} && (*) \\
=& -\frac{1}{2\sigma^2}||x - g(0)||_2^2 + \log \text{Softmax}(\eta(y, z)) + \text{const.} \\
=& \frac{1}{\sigma^2} g(0)^\intercal x + \log \text{Softmax}(\eta(y, z)) + \text{const.} && \text{(Assumption 2.1.A)} \\
=& \frac{1}{\sigma^2} g(0)^\intercal x + \log \frac{e^{\eta(y,z)}}{\sum_z e^{\eta(y,z)}} + \text{const.} \\
=& \frac{1}{\sigma^2} g(0)^\intercal x + \eta(y, z) + \text{const.} && \text{(Assumption 2.1.C)} \\
=& \alpha y^\intercal v + \text{const.} && \text{(Lemma A.1)} \\
=& \alpha(\log q(y|x, z) + \log \sum_{i=1}^{K} e^{v^{(i)}}) + \text{const.} \\
=& \alpha \log q(y|x, z) + \text{const.} && (**)
\end{aligned}
$$

We obtain line $(*)$ for the following reasons: $\log p(y) = $ const. according to Assumption 2.1.B, and $\log p(x, z) = $ const. because only $y$ is variable, $x$ and $z$ are given. We obtained line $(**)$ because given $x$ and $z$, the logits output are fixed. Therefore, $\log \sum_{i=1}^{K} e^{v^{(i)}} = $ const.. Take exponential on both sides of the above equation, we have:

$$p(y|x, z) = \beta q(y|x, z) \tag{19}$$

where $\beta$ is a scale factor. Since both $q(y|x, z)$ and $p(y|x, z)$ are distributions, we have $\sum_y p(y|x, z) = 1$ and $\sum_y q(y|x, z) = 1$. Summing over $y$ on both sides of Equation (19), we have $\beta = 1$. Therefore, we have $q(y|x, z) = p(y|x, z)$. $\qquad\square$

We have proved that CNN with $\sigma_{\text{AdaReLU}}$ and $\sigma_{\text{AdaPool}}$ is the generative classifier derived from the DGM that generates to layer $0$. In fact, we can extend the results to all intermediate layers in the DGM with the following additional assumptions:

**Assumption A.1.** *Each generated layer in the DGM has a constant $\ell_2$ norm: $||g(\ell)||_2^2 = $ const., $\ell = 1, \ldots, L$.*

**Assumption A.2.** *Normalization factor in $p(z|y)$ up to each layer is constant: $\sum_z e^{\eta(y, \{z(j)\}_{j=\ell:L})} = $ const., $\ell = 1, \ldots, L$.*

**Corollary A.1.1.** *Under Assumptions A.1, A.2 and 2.1.B, $p(y|f(\ell), \{z(j)\}_{j=\ell:L})$ in the DGM has the same parametric form as a CNN with $\sigma_{\text{AdaReLU}}$ and $\sigma_{\text{AdaPool}}$ starting at layer $\ell$.*

## A.3 Proof for Proposition 2.1.B

In this section, we provide proofs for Proposition 2.1.B. In the proof, we inherit the notations that we use for proving Theorem 2.1. Without loss of generality, we consider a DGM that has the same architecture as the one we use to prove Theorem 2.1.

**Proposition** 2.1.B. Under Assumption 2.1, MAP estimate of $z(\ell)$ conditioned on $h, y$ and $\{z(j)\}_{j \neq \ell}$ in the DGM is:

$$\hat{z}_R(\ell) = \mathbb{1}(\sigma_{\text{AdaReLU}}(f(\ell)) \geq 0) \tag{20}$$

$$\hat{z}_P(\ell) = \mathbb{1}(g(\ell) \geq 0) \odot \underset{r \times r}{\arg\max}(f(\ell)) + \mathbb{1}(g(\ell) < 0) \odot \underset{r \times r}{\arg\min}(f(\ell)) \tag{21}$$

*Proof.* Without loss of generality, assume that we generate images at a pixel level. In this case, $h = x$. Then we have

$$\underset{z(\ell)}{\arg\max} \log p(z(\ell)|\{z(j)\}_{j \neq \ell}, x, y)$$

$$= \underset{z(\ell)}{\arg\max} \log p(\{z(j)\}_{j=1:L}, x, y)$$

$$= \underset{z(\ell)}{\arg\max} \log p(x|y, \{z(j)\}_{j=1:L}) + \log p(\{z(j)\}_{j=1:L}|y) + \log p(y)$$

$$= \underset{z(\ell)}{\arg\max} \log p(x|y, \{z(j)\}_{j=1:L}) + \eta(y, z) + \text{const.} \qquad \text{(Assumption 2.1.C and 2.1.B)}$$

$$= \underset{z(\ell)}{\arg\max} \frac{1}{\sigma^2} g(0)^{\mathsf{T}} x + \eta(y, z) + \text{const.} \qquad \text{(Assumption 2.1.A)}$$

Using Lemma A.1, the MAP estimate of $z_R(\ell)$ is:

$$\hat{z}_R(\ell) = \underset{z_R(\ell)}{\arg\max}(z_R(\ell) \odot g(\ell))^{\mathsf{T}} f(\ell)$$

$$= \mathbb{1}(\sigma_{\text{AdaReLU}}(f(\ell)) \geq 0)$$

The MAP estimate of $z_P(\ell)$ is:

$$\hat{z}_P(\ell) = \underset{z_P(\ell)}{\arg\max}(z_P(\ell) \odot g(\ell)_\uparrow)^{\mathsf{T}} f(\ell)$$

$$= \mathbb{1}(g(\ell) \geq 0) \odot \underset{r \times r}{\arg\max}(f(\ell)) + \mathbb{1}(g(\ell) < 0) \odot \underset{r \times r}{\arg\min}(f(\ell))$$

$\square$

## A.4 Incorporating instance normalization in the DGM

Inspired by the constant norm assumptions (Assumptions 2.1.A and A.1), we incorporate instance normalization into the DGM. We use $\overline{(\cdot)} = \frac{(\cdot)}{||\cdot||_2}$ to denote instance normalization, and $(\cdot)_n$ to denote layers after instance normalization. In this section, we prove that with instance normalization, CNN is still the generative classifier derived from the DGM. Without loss of generality, we consider a DGM that has the following architecture. We list the corresponding feedforward feature maps on the left column:

|  |  |  |
|---|---|---|
|  |  | $g(0) = W(1)(*^{\mathsf{T}})g_a(1)$ |
| Conv | $f(1) = W(1) * \overline{x}$ | $g_a(1) = g_n(1) \odot z_R(1)$ |
| Norm | $f_n(1) = \overline{f(1)}$ | $g_n(1) = \overline{g(1)}$ |
| ReLU | $f_a(1) = \sigma_{\text{AdaReLU}}(f_n(1) + b(1))$ | $g(1) = W(2)(*^{\mathsf{T}})g_a(2)$ |
| Conv | $f(2) = W(2) * f_a(1)$ | $g_a(2) = g_n(2)_\uparrow \odot z_P(2)$ |
| Norm | $f_n(2) = \overline{f(2)}$ | $g_n(2) = \overline{g(2)}$ |
| Pooling | $f_a(2) = \sigma_{\text{AdaPool}}(f_n(2) + b(2))$ | $g(2) = W(3)^{\mathsf{T}} v$ |
| FC | $v = W(3)f_a(2)$ |  |

**Assumption A.3.** *Feedforward feature maps and feedback feature maps have the same $\ell_2$ norm:*
$$||g(\ell)||_2 = ||f(\ell)||_2, \ell = 1, \ldots, L$$
$$||g(0)||_2 = ||x||_2$$

**Lemma A.2.** *Let $y$ be the label and $x$ be the image. $v$ is the logits output of the CNN that has the same architecture and parameters as the DGM. $g(0)$ is the generated image from the DGM, and $\overline{g(0)}$ is normalized $g(0)$ by $\ell_2$ norm. $\alpha$ is a constant. $\eta(y, z) = \sum_{\ell=1}^{L} \langle b(\ell), z(\ell) \odot g(\ell) \rangle$. Then we have*
$$\alpha y^\intercal v = \overline{g(0)}^\intercal x + \eta(y, z) \tag{22}$$

*Proof.*

$\overline{g(0)}^\intercal x + \eta(y, z)$

$= \{W(1)(*^\intercal)\{g_n(1) \odot z_R(1)\}\}^\intercal \dfrac{x}{||g(0)||_2} + (z_R(1) \odot g(1))^\intercal b(1) + (z_P(2) \odot g(2)_\uparrow)^\intercal b(2)$

$= (z_R(1) \odot g_n(1))^\intercal \{W(1)(*^\intercal)\overline{x}\} + (z_R(1) \odot g(1))^\intercal b(1) + (z_P(2) \odot g(2)_\uparrow)^\intercal b(2)$      (Assumption A.3)

$= g(1)^\intercal \{z_R(1) \odot (f_n(1) + b(1))\} + (z_P(2) \odot g(2)_\uparrow)^\intercal b(2)$      (Assumption A.3)

$= \{W(2)(*^\intercal)\{g_n(2)_\uparrow \odot z_P(2)\}\}^\intercal (z_R(1) \odot f(1)) + (z_P(2) \odot g(2)_\uparrow)^\intercal b(2)$

$= \{g_n(2)_\uparrow \odot z_P(2)\}^\intercal \{W(2) * (z_R(1) \odot f(1))\} + (z_P(2) \odot g(2)_\uparrow)^\intercal b(2)$

$= g(2)^\intercal \{z_P(2) \odot (f_n(2) + b(2))\}$      (Assumption A.3)

$= (W(3)^\intercal y)_\uparrow^\intercal \{z_P(2) \odot f(2)\}$

$= \alpha(W(3)^\intercal y)^\intercal (z_P(2) \odot f(2))_\downarrow$

$= \alpha y^\intercal W(3)(z_P(2) \odot f(2))_\downarrow$

$= \alpha y^\intercal v$

$\square$

**Theorem A.3.** *Under Assumptions A.3 and 2.1.B and 2.1.C, and given a joint distribution $p(h, y, z)$ modeled by the DGM with instance normalization, $p(y|h, z)$ has the same parametric form as a CNN with $\sigma_{AdaReLU}$, $\sigma_{AdaPool}$ and instance normalization.*

*Proof.* The proof of Theorem A.3 is very similar to that of Theorem 2.1 using Lemma A.2. Therefore, we omit the detailed proof here. $\square$

**Remark.** *The instance normalization that we incorporate into the DGM is not the same as the instance normalization that is typically used in image stylization [35]. The conventional instance normalization computes output $y$ from input $x$ as $y = \frac{x - \mu(x)}{\sigma(x)}$, where $\mu$ and $\sigma$ stands for mean and standard deviation respectively. Our instance normalization does not subtract the mean of the input and divides the input by its $\ell_2$ norm to make it have constant $\ell_2$ norm.*

### A.5 CNN-F on ResNet

We can show that CNN-F can be applied to ResNet architecture following similar proofs as above. When there is a skipping connection in the forward pass in ResNet, we also add a skipping connection in the generative feedback. CNN-F on CNN with and without skipping connections are shown in Figure 8.

## B Additional experiment details

### B.1 Standard training on Fashion-MNIST

**Experimental setup** We use the following architecture to train CNN-F: Conv2d(32, $3 \times 3$), Instancenorm, AdaReLU, Conv2d(64, $3 \times 3$), Instancenorm, AdaPool, Reshape, FC(128), AdaReLU, FC(10). The instance normalization layer we use is described in Appendix A.4. All the images are scaled between $[-1, +1]$ before training. We train both CNN and CNN-F with Adam [13], with weight decay of $0.0005$ and learning rate of $0.001$.

We train both CNN and CNN-F for 30 epochs using cross-entropy loss and reconstruction loss at pixel level as listed in Table 1. The coefficient of cross-entropy loss and reconstruction loss is set to be

Figure 8: CNN-F on CNN with and without skipping connections.

1.0 and 0.1 respectively. We use the projected gradient descent (PGD) method to generate adversarial samples within $L_\infty$-norm constraint, and denote the maximum $L_\infty$-norm between adversarial images and clean images as $\epsilon$. The step size in PGD attack is set to be $0.02$. Since we preprocess images to be within range $[-1, +1]$, the values of $\epsilon$ that we report in this paper are half of their actual values to show a relative perturbation strength with respect to range $[0, 1]$.

**Adversarial accuracy against end-to-end attack** Figure 9 shows the results of end-to-end (e2e) attack. CNN-F-5 significantly improves the robustness of CNN. Since attacking the first forward pass is more effective than end-to-end attack, we report the adversarial robustness against the former attack in the main text. There are two reasons for the degraded the effectiveness of end-to-end attack. Since $\sigma_{\text{AdaReLU}}$ and $\sigma_{\text{AdaPool}}$ in the CNN-F are non-differentiable, we need to approximate the gradient during back propagation in the end-to-end attack. Furthermore, to perform the end-to-end attack, we need to back propagate through unrolled CNN-F, which is $k$ times deeper than the corresponding CNN, where $k$ is the number of iterations during evaluation.

(a) Standard training. Testing w/ FGSM.

(b) Standard training. Testing w/ PGD-40.

Figure 9: **Adversarial robustness on Fashion-MNIST against end-to-end attack.** CNN-F-$k$ stands for CNN-F trained with $k$ iterations; PGD-$c$ stands for a PGD attack with $c$ steps. CNN-F achieves higher accuracy on MNIST than CNN for under both standard training and adversarial training. Each accuracy is averaged over 4 runs and the error bar indicates standard deviation.

## B.2 Adversarial training

**Fashion-MNIST** On Fashion-MNIST, we use the following architecture: Conv2d(16, $1\times 1$), Instancenorm, AdaReLU, Conv2d(32, $3\times 3$), Instancenorm, AdaReLU, Conv2d(32, $3\times 3$), Instancenorm, AdaReLU, AdaPool($2\times 2$), Conv2d(64, $3\times 3$), Instancenorm, AdaReLU, AdaPool($2\times 2$), Reshape, FC(1000), AdaReLU, FC(128), AdaReLU, FC(10). The intermediate reconstruction losses are added at the two layers before AdaPool. The coefficients of adversarial sample cross-entropy losses and reconstruction losses are set to 1.0 and 0.1, respectively. We scaled the input images to [-1,+1]. We trained with PGD-7 attack with step size 0.071. We report half of the actual $\epsilon$ values in the paper to show a relative perturbation strength with respect to range $[0, 1]$. To train the models, we use SGD optimizer with learning rate of 0.05, weight decay of 0.0005, momentum of 0.9 and gradient clipping with magnitude of 0.5. The batch size is set to be 256. We use polynomial learning rate scheduling with power of 0.9. We trained the CNN-F models with one iteration for 200 epochs using the following hyper-parameters: online update step size 0.1, ind 2 (using two convolutional layers to encode images to feature space), clean coefficients 0.1.

**CIFAR-10** On CIFAR-10, we use Wide ResNet (WRN) [39] with depth 40 and width 2. The WRN-40-2 architecture consists of 3 network blocks, and each of them consists of 3 basic blocks with 2 convolutional layers. The intermediate reconstruction losses are added at the layers after every network block. The coefficients of adversarial sample cross-entropy losses and reconstruction losses are set to 1.0 and 0.1, respectively. We scaled the input images to [-1,+1]. We trained with PGD-7 attack with step size 0.02. We report half of the actual $\epsilon$ values in the paper to show a relative perturbation strength with respect to range $[0, 1]$. To train the models, we use SGD optimizer with learning rate of 0.05, weight decay of 0.0005, momentum of 0.9 and gradient clipping with magnitude of 0.5. The batch size is set to be 256. We use polynomial learning rate scheduling with power of 0.9. We trained the models for 500 epochs with 2 iterations. For the results in Table 3, we trained the models using the following hyper-parameters: online update step size 0.1, ind 5, clean coefficients 0.05. In addition, we perform an ablation study on the influence of hyper-parameters.

**Which layer to reconstruct to?** The feature space to reconstruct to in the generative feedback influences the robustness performance of CNN-F. Table 4 list the adversarial accuracy of CNN-F with different ind configuration, where "ind" stands for the index of the basic block we reconstruct to in the first network block. For instance, ind=3 means that we use all the convolutional layers before and including the third basic block to encode the input image to the feature space. Note that CNN-F models are trained with two iterations, online update step size 0.1, and clean cross-entropy loss coefficient 0.05.

Table 4: Adversarial accuracy on CIFAR-10 over 3 runs. $\epsilon = 8/255$.

|  | Clean | PGD (first) | PGD (e2e) | SPSA (first) | SPSA (e2e) | Transfer | Min |
|---|---|---|---|---|---|---|---|
| CNN-F (ind=3, last) | $78.94 \pm 0.16$ | $46.03 \pm 0.43$ | $60.48 \pm 0.66$ | $68.43 \pm 0.45$ | $64.14 \pm 0.99$ | $65.01 \pm 0.65$ | $46.03 \pm 0.43$ |
| CNN-F (ind=4, last) | $78.69 \pm 0.57$ | $47.97 \pm 0.65$ | $56.40 \pm 2.37$ | $69.90 \pm 2.04$ | $58.75 \pm 3.80$ | $65.53 \pm 0.85$ | $47.97 \pm 0.65$ |
| CNN-F (ind=5, last) | $78.68 \pm 1.33$ | $\mathbf{48.90 \pm 1.30}$ | $49.35 \pm 2.55$ | $68.75 \pm 1.90$ | $51.46 \pm 3.22$ | $66.19 \pm 1.37$ | $\mathbf{48.90 \pm 1.30}$ |
| CNN-F (ind=3, avg) | $79.89 \pm 0.26$ | $45.61 \pm 0.33$ | $\mathbf{67.44 \pm 0.31}$ | $68.75 \pm 0.66$ | $\mathbf{70.15 \pm 2.21}$ | $64.85 \pm 0.22$ | $45.61 \pm 0.33$ |
| CNN-F (ind=4, avg) | $80.07 \pm 0.52$ | $47.03 \pm 0.52$ | $63.59 \pm 1.62$ | $70.42 \pm 1.42$ | $65.63 \pm 1.09$ | $65.92 \pm 0.91$ | $47.03 \pm 0.52$ |
| CNN-F (ind=5, avg) | $\mathbf{80.27 \pm 0.69}$ | $48.72 \pm 0.64$ | $55.02 \pm 1.91$ | $\mathbf{71.56 \pm 2.03}$ | $58.83 \pm 3.72$ | $\mathbf{67.09 \pm 0.68}$ | $48.72 \pm 0.64$ |

**Cross-entropy loss coefficient on clean images** We find that a larger coefficient of the cross-entropy loss on clean images tends to produce better end-to-end attack accuracy even though sacrificing the first attack accuracy a bit (Table 5). When the attacker does not have access to intermediate output of CNN-F, only end-to-end attack is possible. Therefore, one may prefer models with higher accuracy against end-to-end attack. We use cc to denote the coefficients on clean cross-entropy. Note that CNN-F models are trained with one iteration, online update step size 0.1, ind 5.

Table 5: Adversarial accuracy on CIFAR-10 over 3 runs. $\epsilon = 8/255$.

|  | Clean | PGD (first) | PGD (e2e) | SPSA (first) | SPSA (e2e) | Transfer | Min |
|---|---|---|---|---|---|---|---|
| CNN-F (cc=0.5, last) | $82.14 \pm 0.07$ | $45.98 \pm 0.79$ | $59.16 \pm 0.99$ | $72.13 \pm 2.99$ | $59.53 \pm 2.92$ | $67.27 \pm 0.35$ | $45.98 \pm 0.79$ |
| CNN-F (cc=0.1, last) | $78.19 \pm 0.60$ | $47.65 \pm 1.72$ | $56.93 \pm 9.20$ | $66.51 \pm 1.10$ | $61.25 \pm 4.23$ | $64.93 \pm 0.70$ | $47.65 \pm 1.72$ |
| CNN-F (cc=0.05, last) | $78.68 \pm 1.33$ | $\mathbf{48.90 \pm 1.30}$ | $49.35 \pm 2.55$ | $68.75 \pm 1.90$ | $51.46 \pm 3.22$ | $66.19 \pm 1.37$ | $\mathbf{48.90 \pm 1.30}$ |
| CNN-F (cc=0.5, avg) | $\mathbf{83.15 \pm 0.29}$ | $44.60 \pm 0.53$ | $\mathbf{68.76 \pm 1.04}$ | $72.34 \pm 3.54$ | $\mathbf{68.80 \pm 1.18}$ | $67.53 \pm 0.48$ | $44.60 \pm 0.53$ |
| CNN-F (cc=0.1, avg) | $80.06 \pm 0.65$ | $46.77 \pm 1.38$ | $63.43 \pm 7.77$ | $69.11 \pm 0.77$ | $66.25 \pm 4.40$ | $65.56 \pm 1.08$ | $46.77 \pm 1.38$ |
| CNN-F (cc=0.05, avg) | $80.27 \pm 0.69$ | $48.72 \pm 0.64$ | $55.02 \pm 1.91$ | $71.56 \pm 2.03$ | $58.83 \pm 3.72$ | $67.09 \pm 0.68$ | $48.72 \pm 0.64$ |