[Reviews · NeurIPS 2020]

Review 1

Summary and Contributions: The authors develop a new model, CNN-F, that enforces self consistency of the internal representations within a Bayesian framework via recurrent generative feedback connections added to a CNN. The authors demonstrate that this model provides better predictions of visual neural activity and better adversarial robustness than a standard CNN. They also show that the generative feedback can restore perturbed images.

Strengths: The authors port the interesting cognitive ideas about self-consistency to CNNs using a mathematically rigorous framework. I enjoyed reading this paper because it did not focus merely on small adjustments to reach SOTA but instead presented a well-motivated and creative framework which seems fairly significant and novel. One key result the authors present from their model is that it predicts visual neural responses slightly better, despite slightly worse performance on ImageNet, when compared to a standard CNN. This finding is interesting, although the improvements for neural response prediction do not seem very large. I would also have liked to see some of the "raw data" instead of just the Pearsons' r summary over all neurons. The more striking findings in this paper concern the significantly better adversarial robustness of the CNN-F model and its ability to restore perturbed images. The improvement over standard CNNs in response to adversarial attacks is particularly impressive (and important for the field). The results all fit together nicely to provide evidence that the CNN-F is operating as the authors intended, by imposing self-consistency on the internal representations. Overall, the authors present a well-motivated novel model with interesting and internally consistent results. This paper is extremely relevant to the NeurIPS community.

Weaknesses: One major weakness of this paper is the lack of clarity, especially in the presentation of the details of the CNN-F. I review that in the relevant section below. Another weakness is the lack of other comparison models/methods. The authors always compare CNN-F to CNN. I would like to see more comparisons in Figures 5, 6, and 7. How does the neural prediction performance of CNN-F compare to CORnet? How does the adversarial robustness compare to other networks specifically created to be adversarially robust? Is the generative feedback especially good at restoring perturbed images when compared to alternatives? One small weakness is the confusing and inaccurate description of predictive coding in lines 41-46. Predictive coding applies to a much broader set of models than the specific version described here. It is definitely not necessarily a "formulation of hierarchical Bayesian inference that assumes Gaussian distributions." This section also seems to imply that predictive coding has been figured out - it very much has not been! "Recurrent feedback pathways to perform Bayesian inference" is just one possible implementation for predictive coding. For example, ascending information about other sensory motion and self motion cues could also drive predictive processing in a non-recurrent manner (see Schneider et al Nature Letters 2018). --- UPDATE AFTER AUTHOR REBUTTAL --- Thanks to the authors for their responses. I am keeping my score as is (recommending acceptance) but second R4s call for a clear systematic comparison to other models on Brain-Score.

Correctness: As far as I could understand all the specific methods of the model (see Clarity section), the methodology seems correct.

Clarity: In general, I found this paper difficult to follow. In particular, the CNN-F model was more difficult to understand that it should have/could have been. One improvement would be to have a better schematic of the model: Figure 2 was unhelpful to me without more explanation, especially as the first relevant figure, and Figure 3 could definitely be improved. Another would be to much more explicitly address how h/f/g etc relate to each other. After several read throughs, I think I arrived at a better understanding of the model but it should not take that many for a reader familiar with this type of modeling. Introduction: I also found the discrete contributions sections in the introduction confusing as they overly obscured how the contributions linked together and what the authors had actually done. Lines 32-34 also do not make sense. Sections 3/4/5/6: These sections were all much clearer and fairly well-written. Overall: needs a careful read-through for typos/weird capitalization

Relation to Prior Work: The authors clearly discuss related work, although I would have preferred this section to be much earlier in the paper.

Reproducibility: Yes

Additional Feedback:


Review 2

Summary and Contributions: Inspired by current hypothesis in neurosciences (Probabilistic brain, predictive coding), this work proposes to add generative feedback connections to classical CNN architecture. The proposed architecture (CNN-F) is a combination of a CNN and a Deconvolutional Generative Model (DGM). The authors show that the trained model (CNN-F) marginally provides a better account for primate vision neural responses than standard CNN. Then, they show that CNN-F is more robust to adversarial attacks and is able to restore perturbed images.

Strengths: * the work partially closes the gab between CNN and human vision performances * the improvement comes from combining feedback connections and "self-consistency"

Weaknesses: I do not see any weaknesses apart from minor formatting remarks or lack of mathematical rigor.

Correctness: Except from the 2 following remarks I do not see any incorrect claims or claims: * logistic regression is not derived from Gaussian naive Bayes (GNB). The former is a discriminative model that directly defines a posterior distribution while the later is a generative model that requires Bayes rule to compute the posterior (which is different from the logistic proba). * Theorem 2.1 is not rigorously stated

Clarity: A minor remark: repetition is a way to teach but it's sometimes too much in this paper eg the name of the model is repeated at least 3 times (l. 10, l. 65, l. 83 and l. 127) and it's not a essential information. The text in the figure is too small.

Relation to Prior Work: Prior work is sufficiently cited and discussed. Contributions of this paper are clear.

Reproducibility: Yes

Additional Feedback: Detailed remarks: * l. 3: I think, computational neuroscientist have shifted towards using probabilistic brain instead of Bayesian brain. * l. 86: an image & is a one-hot * l. 90: layer L to l ? & Do not start a sentence with a mathematical symbol. * l. 94: Do not start a sentence with a mathematical symbol. * l. 132, eq 2 : what operation is the circle with a dot inside ? What operation is *^T ? * Theorem 2.1: The notion of generative classifier is not defined. The notion that it is derived from a model such as DGM is also not defined... I think I understand that A and C are important for the implementation but what would be the role (if made understandable) of B ? * l. 175: \hat h_k ? * Figure 5: are there error bars ? Any cross-validation ? * l. 211: what is BPDA ? * Figure 6: error bars shouldn't be standard deviation but 2 or 3 times the standard error of the mean. To improve readability the bar plot can be cut above 0 (the min of the y axis). Questions: * Does self-consistency always achievable ? * Is it possible to design an attack that is specific to CNN-F * CNN-F is able to restore perturbed images, would is also wrongly restore image in a way that is compatible with some visual illusion such as Kanizsa triangle (ie generating somehow fake contours) ? ### Post author response ### After reading author response and discussion with the other reviewers I decided to maintain my score.


Review 3

Summary and Contributions: This paper proposes a probabilistic model for classification and input denoising. A generative feedback model is first defined, consisting of a hierarchy of Bernoulli latent variables, linked with transposed convolutions and sigmoidal nonlinearities. From this model, a forward neural network is obtained, with convolutions, rectifiers and maxpooling. The two pathways interact at every layer, since the forward nonlinearities are modulated by the feedback. An alternating inference algorithm is proposed. Experiments demonstrate that the model is more robust to adversarial perturbations and can better predict primate V4 and IT cortex neural activity than a baseline feedforward model. UPDATE: I maintain my decision to recommend acceptance after the discussion phase.

Strengths: - Well grounded in Bayesian brain theories while remaining creative. - Elegant linking of forward and feedback pathways. - Image denoising, robustness to adversarial attacks and a better 'BrainScore' are excellent demonstrations of the method.

Weaknesses: - Classification performance consistently and quite significantly decreases with the proposed method. The paper would improve if some discussion on this was provided. - Related to the prior point, results do not clearly demonstrate that the model can be trained from scratch to good performance. The ImageNet experiment starts from a pretrained model and the Fashion-MNIST accuracy is not impressive for the architecture. - Restriction to DGM model seems somewhat arbitrary, as the ideas appear to be more general in scope.

Correctness: - It seems important to discuss Assumption A1-3 (found in the SM) which arise in Theorem 2.1 in the main text. - The authors introduce a special 'instance normalization' step in their model. Is the baseline CNN also modified to include such a step? If not, it would be important to control for this in the experiments, to show that the advantages do not come from this step.

Clarity: To summarize my position here, I enjoyed reading the motivation parts of the paper, but I struggled with the presentation of the model, Section 2. For example, layers were often not explicitly denoted. Right at the start, it took me a long time to understand Fig. 3, and what h was. Only with the help of the SM (A.1) I finally understood what was meant by $h$, which turned out to be quite simple, just that one could consider generating only up to a certain convolutional layer $k$ instead of all the way to the 0-th layer. In retrospect this was right there in the text, but I couldn't parse it. Stemming from the notation problem above, early on I got confused with regards to weight tying between the parameters of f and g. Note that in Fig. 3, the same W reappears in f and g (good; weight sharing which relates f to the generative model) but then in multiple layers (confusing, sharing across layers?).

Relation to Prior Work: I found the citations and positioning of the paper appropriate.

Reproducibility: Yes

Additional Feedback: - The paper has some typos throughout; please correct them. - It would be nice to discuss the inference algorithm derived by the authors in the light of the predictive coding/processing framework, which has greatly influenced neuroscience thinking. Do the authors see a way of implementing (at a high description level) their alternating procedure by passing predictions and prediction errors throughout the hierarchy? Does the model require bidirectional propagation of prediction errors (of a different kind)? - Could the authors discuss if they envision a way to sample from the model and go beyond point estimates, towards a 'proper' Bayesian brain?


Review 4

Summary and Contributions: This paper describes a generative hierarchical recurrent model which can be trained to solve object recognition tasks. The model is inspired by notions of predictive coding, and implements "explaining away". The model is validated on imagenet, brainscore, adversarial robustness, and some qualitative examples of image restoration through feedback on a toy dataset. --- Update --- I really struggle with this paper. I like the ideas, and I am delighted to see a successful deep generative feedback network. However, I think the evaluations are not convincing of the model's effectiveness above and beyond existing feedforward or recurrent networks. In particular, the Brain-Score evaluations are very imprecise. The author's rebuttal's contained a figure showing that a standard VGG-16 outperforms this model at explaining V4, when the opposite result was reported in the initial draft. The rebuttal showed that the model was better than VGG-16 at explaining IT, but VGG-16 is not state-of-the-art at explaining IT. For the final version of this paper, please provide a cleaner systematic comparison of this model to others on Brain-Score. Right now it looks like cherry-picking (at best). The other reviewers pointed out that I was overly harsh in evaluating the adversarial robustness. They have a good point and so I will raise my score.

Strengths: The model is quite interesting, and is likely of interest to cognitive and neuroscientists, particularly for reconciling the performance of deep learning models on computer vision benchmarks with theories of predictive coding from neuroscience and cognitive science. The proof for self-consistency is also interesting, and could potentially be of interest to people studying neural circuits.

Weaknesses: --Introduction Figure 1 does nothing for me. The input image arives somewhere in PFC, then there's FF connections from there to an internal model? and FB connections + the category guess comes from v1? --Model description Are adapool and adarelu your creations? If not, how is the model different than the work that introduced those functions? Are these necessary to make the model work? What happens to model training when you use only one or two of the three losses you describe for training? How does that affect performance? Can you construct analagous versions of these losses for the feedforward models that you're comparing to, in order to ensure that the results you find can't simply be explained by introducing more constraints in the objective function? How many steps of recurrence/inference-time optimization does it take for your model to converge to a solution? --Neural comparison I have three issues with the brainscore comparison in Figure 5. First, the underlying princple for that work is that better performance on ImageNet equates to better performance on Brainscore. Here you show the opposite. This is not always true, of course (the Brainscore authors' own model is not especially good at classification), but it needs to be discussed here. Second, you outperform VGG-16 on brainscore, but it seems like you have much more to your model than a VGG16 (perhaps I misunderstood something...). You need to construct a version of VGG16 which is purely feedforward, but as close as possible to your model as you can make it, including adarelu, adapool, some version of the multiple losses. Otherwise I cannot tell *why* your model is explaining neural activity better than the VGG16. Third, I'm not sure where the predictivity scores here come from. I checked the website and VGG-16 is 0.627 for V4 and 0.513 for IT. Furthermore, while VGG16 does seem to be SOTA for V4, it is not for IT, and a more appropriate comparison would be to the leading models for both of these brain areas. --Adversarial robustness I realize I may be pushing against a standard in the field, but the way papers -- including this one -- are justifying adversarial robustness makes things impossible to evaluate. When you trade off performance on the standard dataset (Figure 6, "0.0") for improved adversarial robustness, it is difficult to understand whether the robustness emerges from worse classification performance or some other means. Could similar results be achieved through some other regularization? It is hard to pinpoint what makes this method stand out. In the absence of improving standard and perturbed datasets, perhaps the best solution is to systematically compare against other adversarially robust models? Another issue I have is that the experiments were done on toy datasets. Why not stick to ImageNet? --Image restoration. This set of experiments is very difficult to interpret. I would suggest figuring out some way to make this falsifiable. I don't really even see any "filling in" or some other image manipulation that would be consistent with explaining away.

Correctness: See Weaknesses for questions I have about the brainscore evaluation. Otherwise methods seem correct.

Clarity: It is OK. I'd appreciate more of a deep dive into the model, maybe move some of the SI to the main text? Figure 1 must be changed.

Relation to Prior Work: Yes.

Reproducibility: Yes

Additional Feedback:

[Author Response · NeurIPS 2020]

We thank the reviewers for their thoughtful feedback. We are encouraged that all the reviewers found our work to
be creative in porting predictive coding and Bayesian brain theories in neuroscience to deep learning models using a
mathematically rigorous framework. We want to assure the reviewers that we will heed their advice such as making the
figures more informative and interpretable as well as stating the theorems rigorously. We want to thank **R2** and **R3** for
their insightful questions which actually align with some future works that we have envisioned based on this paper. We
will add a session in the revised paper to answer them. Due to space limit, we address the main concerns here.

*Model description* **R4 Analogous version of AdaReLU, AdaPool, and losses in a purely feedforward model** We
would like to clarify this potential misunderstanding. The fundamental difference between a CNN-F and a CNN is recur-
rent generative feedback; the CNN-F's initial feedforward step is equivalent to a feedforward CNN. The CNN-F's feed-
back and subsequent feedforward steps use the adaptive layers and multiple losses. AdaReLU and AdaPool are our own
creations and are required to perform Bayesian inference. They modulate the top-down feedback and change the ReLU
and MaxPool units in the feedforward pathway. Since AdaReLU, AdaPool, and generative losses (reconstruction and con-
ditional latent likelihood loss) all need to act on feedback signals, there are no analogies in a purely feedforward model.

**R4 How do losses affect performance?** We included the ablation study in Appendix (line 522).

**R4 Convergence of CNN-F** Empirically, we find that 5 iterations lead to a stable solution. We
compute the mean square difference between successive iterations to show convergence. The
changes in reconstructed images $x$ and logits $y$ are shown in Figure 1. As expected, clean
images converge faster than adversarial images.

**R3 Restriction to the DGM model** Thanks for noticing that the ideas can be more general.
However, given CNN as the architecture for classification, DGM is required to enforce Bayes
rule. For other architectures, we can use self-consistency to define iterative inference accordingly.

*Neural comparison* **R1 Comparison with CORnet** CORnet proposes recurrent connections within each cortical
area while CNN-F proposes feedback across areas. We included the CORnet-S and CORnet-Z neural similarity scores.
The CNN-F with the VGG-16 architecture outperforms the CORnet-S, and CORnet-Z in V4 and IT neural similarity.

**R4 The predictivity scores do not match brain-score.org.** Due to com-
putational limitations, we were previously not able to perform an extensive
hyperparameter search on which layer to predict neural activity for V4 and
IT. We ran this study with the results in Figure 2.

**Neural predictivity discussion** (**R4**) According to Schrimpf et al. (2018),
the correlation between accuracy and Brain-Score becomes insignificant
models with ImageNet top-1 accuracies greater than 70%. This can be
expected as V4 and IT perform other roles besides strictly object classification. As for the decreased classification
performance of CNN-F, please also refer to a related question of **R3** on line 50.

*Adversarial robustness (AR)* **R1**, **R4 Compare against adversarially robust models** The arguably most established
method to improve the AR of neural networks is adversarial training (AT) (Madry et al., 2018). In the paper, we show
that CNN-F can further improve the robustness of adversarially trained CNNs (line 229).

**R4 Trade off between clean and adversarial accuracy** We fully understand your concern. Trade-off between
robustness and accuracy is still an active research area (Hongyang Zhang et al., 2019, Yao-Yuan Yang et al., 2020). In
fact, we think that CNN-F provides a new lens to analyze this trade-off. We find that more iterations are needed for
adversarial (harder) images and less iterations are needed for clean (easier images) (Fig 6. (f)). By varying the number
of iterations, a CNN-F can fit different function classes to clean and adversarial images while achieving high accuracy.

**R4 CNN-F v.s. regularization** Indeed, regularization is a useful way to improve AR. Yuxin Wen et al. (2020) shows
that AT regularizes NNs to concentrate samples around decision boundaries. Inspired by predictive coding, we propose
self-consistency as a different mechanism for robustness. Our approach is compatible with other regularization such as
AT. We show that feedback improves upon AT (Fig 6. (d)) and also generalizes better to unseen attacks (Fig 6. (e)).

**R4 Why not stick to ImageNet for AR** MNIST, Fashion-MNIST and CIFAR-10 are standard benchmarks across the
AR community and haven't been solved yet. We also conducted experiments on CIFAR-10 and found that CNN-F has
higher adversarial accuracy than CNN in later iterations and maintains as high clean accuracy in earlier iterations. Due
to the space limit, we omit the results here but will put into the revised paper.

*Training difficulties (**R3**)* **Classification performance** There are three reasons for the decreased classification
performance. First, generative classifier has higher asymptotic error compared to discriminative classifier when
there is a mismatch between data and model (Ng et al., 2002). Second, it is harder to optimize a recur-
rent model like CNN-F than a feedforward model like CNN. Third, CNN-F achieves higher accuracy for
clean images in early iterations (Fig. 6 (f), $\epsilon = 0.0$), indicating shorter processing time for easy images.

**Training from scratch** We need to use Instance Normalization (IN) to train CNN-F from
scratch. We also used IN in CNN for fair comparison. The accuracies are in Table 1. For the
pretrained model, we were not using any layer normalization in both CNN and CNN-F.

| | Top-1 | Top-5 |
|---|---|---|
| CNN | 58.85 | 77.50 |
| CNN-F | 56.68 | 74.00 |

[Meta-Review · NeurIPS 2020]

Three knowledgeable referees support acceptance for the contributions, notably for a proposed approach to extend CNNs with generative feedback, while one reviewer supports (marginal) reject. This paper was extensively discussed post-rebuttal — especially in light of the fact that the initial evaluation on brainscore appeared to have been flawed and that the results on brainscore have not just changed quantitatively but also qualitatively. I also agree with R4 that the overall evaluation is not particularly compelling as a general model of object recognition (see R4 points) as opposed to maybe a narrower approach to build robustness to adversarial attacks. Overall, there appears to be sufficient support because of the novelty of the idea to accept this paper but all reviewers agreed that the quantitative evaluation on brainscore needs to be fixed and claims revised accordingly.